# Untapped Neuroimaging Tools for Neuro-Oncology: Connectomics and Spatial Transcriptomics

**DOI:** 10.3390/cancers14030464

**Published:** 2022-01-18

**Authors:** Jurgen Germann, Gelareh Zadeh, Alireza Mansouri, Walter Kucharczyk, Andres M. Lozano, Alexandre Boutet

**Affiliations:** 1Department of Surgery, Division of Neurosurgery, University of Toronto, Toronto, ON M5T 2S8, Canada; Gelareh.Zadeh@uhn.ca (G.Z.); Andres.Lozano@uhnresearch.ca (A.M.L.); 2MacFeeters Hamilton Neuro-Oncology Program, Princess Margaret Cancer Centre, Toronto, ON M5G 1L7, Canada; 3University Health Network and University of Toronto, Toronto, ON M5G 2C4, Canada; 4Princess Margaret Cancer Centre, University Health Network, Toronto, ON M5G 2C1, Canada; 5Krembil Brain Institute, Toronto, ON M5T 2S8, Canada; 6Department of Neurosurgery, Penn State Hershey Medical Center, Penn State University, Hershey, PA 17033, USA; amansouri@pennstatehealth.psu.edu; 7Joint Department of Medical Imaging, University of Toronto, Toronto, ON M5T 1W7, Canada; w.kucharczyk@utoronto.ca; 8Center for Advancing Neurotechnological Innovation to Application (CRANIA), Toronto, ON M5T 1M8, Canada

**Keywords:** neuro-oncology, neuroimaging, MRI, normative analysis, connectomics, imaging transcriptomics

## Abstract

**Simple Summary:**

Brain imaging, specifically magnetic resonance imaging (MRI), plays a key role in the clinical and research aspects of neuro-oncology. Novel neuroimaging techniques enable the transformation of a brain MRI into a so-called average brain. This allows projects using already acquired brain MRIs to perform group analyses and draw conclusions. Once the data are in this average brain, several types of analyses can be performed. For example, determining the most vulnerable locations for certain tumor types or perhaps even the underlying circuitry and gene expression that might cause predisposition to tumor growth. This information may further our understanding of tumor behavior, leading to better patient counseling, surgery timing, and treatment monitoring.

**Abstract:**

Neuro-oncology research is broad and includes several branches, one of which is neuroimaging. Magnetic resonance imaging (MRI) is instrumental for the diagnosis and treatment monitoring of patients with brain tumors. Most commonly, structural and perfusion MRI sequences are acquired to characterize tumors and understand their behaviors. Thanks to technological advances, structural brain MRI can now be transformed into a so-called average brain accounting for individual morphological differences, which enables retrospective group analysis. These normative analyses are uncommonly used in neuro-oncology research. Once the data have been normalized, voxel-wise analyses and spatial mapping can be performed. Additionally, investigations of underlying connectomics can be performed using functional and structural templates. Additionally, a recently available template of spatial transcriptomics has enabled the assessment of associated gene expression. The few published normative analyses have shown relationships between tumor characteristics and spatial localization, as well as insights into the circuitry associated with epileptogenic tumors and depression after cingulate tumor resection. The wide breadth of possibilities with normative analyses remain largely unexplored, specifically in terms of connectomics and imaging transcriptomics. We provide a framework for performing normative analyses in oncology while also highlighting their limitations. Normative analyses are an opportunity to address neuro-oncology questions from a different perspective.

## 1. Introduction

Neuroimaging plays a crucial role in neuro-oncology research. At present, magnetic resonance imaging (MRI) is a key tool for the diagnosis and treatment monitoring of brain tumors. Most commonly, high spatial resolution structural images, such as T1 and T2-weighted sequences with and without gadolinium contrast agents, are acquired [1]. Perfusion imaging has also become a mainstay of neuro-oncology imaging protocols in the past decade, in which it has been used for brain tumor characterization as well as for the evaluation of treatment response and disease progression [2]. Generally, neuro-oncology-based analyses investigate tumors and their treatment in terms of signal characteristics across patients. Thanks to recent technological advances, group analyses can be performed in a so-called average brain [3,4]. However, while they have demonstrated their usefulness in other research fields, these analyses are uncommonly used in neuro-oncology.

Neuroimaging advances have enabled precise transformation of a patient’s brain onto an average brain template. Sophisticated computer algorithms use non-rigid methods to transform an individual brain to the brain template while accounting for anatomical differences [3,4]. Within the template, each voxel has a distinct *address*, which allows comparisons across subjects with varied brain morphologies, also referred to as analysis in normative space. Subsequently, voxel-based group statistics, functional and structural connectomics, and imaging transcriptomics analyses can be performed [5,6]. Importantly, these analyses only require high spatial resolution structural MRI, which are routinely acquired in clinical neuro-oncology.

Brain lesions, including tumors, can have local and distant clinical and histopathological effects that can be caused, for example, by local mass effects and invasion as well as connectomal diaschisis, respectively. In this perspective, we aim to provide an overview of normative voxel-based, connectomics, and imaging transcriptomics group analyses, which are currently uncommonly used in neuro-oncology research. We will first provide a technical summary to explain these methods. Then, we will discuss the few published neuro-oncology examples, as well as several potential avenues for these techniques. We will end by providing a practical framework to use these methods while also highlighting their limitations.

## 2. Advanced Neuroimaging Analyses Using Normative Brain Templates

There is considerable variability between individual human brains. This variability poses a challenge when comparing and communicating findings related to normal and altered brain anatomy and function. To overcome this limitation, a brain reference space, based on many individual brains, has been proposed: the Montreal Neurological Institute (MNI) template [4,7,8,9]. This common space was widely adapted in the late 1990s [10]. It serves as the standard reference space for the analysis and reporting of results. A multitude of multimodal publicly available atlases have been accurately registered in this standard space.

To be able to use the tools and atlases available, individual brains have to be projected onto the template space. This normalization step transforms a brain to match the template brain and typically first involves rotation, translation, scaling, and shearing. These are linear operations applied to all voxels and this is usually referred to as linear registration/transformation. In the next step, voxels are non-uniformly locally warped to better match the template brain anatomy. This non-linear registration maximizes anatomical correspondence between the individual brain and the template [11]. Once all brains have been transformed into common space, one can now compare and make statistical inferences about local anatomical differences [12] or map spatial characteristics of tumors and their clinical attributes [13] (Figure 1). At this point, voxel-based analyses can be performed.

### 2.1. Connectomics

The brain is a complex system of a multitude of distinct areas that are interconnected [14,15,16,17,18]. The networks formed by these groups of brain elements, as well as the interaction within and between each network, form the basis of behaviors. The connectome is key for understanding how the brain works. Over the last years, multiple studies have demonstrated that neurological and psychiatric symptoms can be mapped to a common distributed brain network [19,20,21,22,23,24,25,26,27,28,29,30]. The brain connectome can be assessed using different MRI acquisitions: (1) diffusion-weighted MRI (dMRI) to estimate the structural connectivity using the directionality of water diffusion to evaluate tracts and projections [31]; (2) resting state functional MRI (rsfMRI) to estimate functional connectivity making use of the low-frequency blood oxygen level-dependent (BOLD) fluctuations found in brain regions that are functionally related to each other [32]. The functional (i.e., rsfMRI) and structural (i.e., dMRI tractography) connectomes can be studied using normative, atlas-based connectome data assembled using high-quality acquisitions from a large number of subjects [5,26,33]. The use of these normative connectomes allows the investigation of brain-wide circuits implicated in clinical symptoms in the absence of patient-specific rsfMRI or dMRI acquisitions [24,29,34,35,36] (Figure 1). For example, if tumors are found to cause a specific cognitive impairment, one could use the normative functional connectome to map the key implicated regions and optimize patient counseling and surgical timing of subsequent tumors with similar characteristics.

### 2.2. Imaging Transcriptomics

These normative functional and structural connectomes have been highly valuable resources for studying normal and abnormal brain processes. Spatially resolved gene expression data provide yet another unique avenue to investigate the relationship between brain circuits implicated in various behaviors and diseases, and cell organization and underlying molecular characteristics [6,37]. The Allen Human Brain atlas provides spatially resolved gene expression in a standardized template space [38,39,40]. It is an online open access atlas of whole-brain microarray gene expression data (https://help.brain-map.org/display/humanbrain/Documentation, accessed on 1 December 2021). Using post-mortem tissue samples from six donors (age range 24–57 years, 1 female), over 20,000 genes were sampled across 3702 distinct sites creating an anatomically comprehensive spatial gene expression assay of the human brain [40]. This normative gene expression atlas allows one to investigate the molecular and cellular mechanisms associated with a spatial pattern or brain location identified in an independent neuroimaging analysis (Figure 1). For example, a study by Zheng and colleagues used normative connectomics and transcriptomics to investigate determinants of disease propagation in Parkinson’s disease [41] and recent work by Mandal and colleagues demonstrated that the spatial distribution of gliomas is related to the local normative expression of genes associated with chromatin organization and synaptic signaling [41,42].

## 3. Neuro-Oncology Applications

The neuro-oncology literature contains a few examples of normative analyses; however, several potential applications remain unexplored, specifically connectomics and imaging transcriptomics.

The majority of publications have used normative voxel-wise mapping to investigate spatial patterns of tumors. Spatial mapping has shown that glioblastomas tend to occur in different locations according to their molecular subtypes [43]. Similar analyses identified brain areas favored by lung cancer metastases, as well as a relationship between their spatial distribution and epidermal growth factor receptor mutation status [44,45]. Pediatric post-surgical cerebellar lesion mapping was used to identify structures most strongly associated with cognitive affective syndrome [46].

There is a paucity of neuro-oncology studies that have explored the normative functional connectome. One study found that the underlying normative circuitry of a group of spatially distinct mass lesions could explain the development of medically refractory epilepsy [36]. Following further validation, this approach could be used for patient counseling and optimizing surgical timing for lesions in locations involving epileptogenic networks. This type of analysis could also assess other tumor characteristics, such as preferential localization to specific functional networks based on tumor histology and genetics. There is also a case report of post-operative depression after cingulate low-grade glioma resection, in which functional connectomics suggests that the surgical corridor, rather than the tumor resection bed, had greater overlap with depression-related networks [47]. Furthermore, functional connectomics has been used to investigate the key functional networks underlying cognitive affective syndrome after cerebellar lesion resection [46]. Normative functional connectomic analyses are most appropriate when investigating distinct neurological manifestations from lesions in anatomically disparate locations. This analysis can be used to correlate symptoms and lesions, and may be a valuable tool for pre-operative patient counseling and selecting the optimal surgical approach.

There are very few publications taking advantage of normative structural connectomics and imaging transcriptomics. There is evidence that certain types of tumors exhibit preferential growth along white matter fiber directions [48,49]. Tumor growth patterns could be further assessed with large-scale data using normative structural connectomics. Two recently published reviews also suggest the application of brain connectomics for glioma surgery [50,51]. Similarly, understanding tumor behaviors and associated symptoms might require a whole-brain approach [52]. Rather than considering tumors with traditional lesion mapping, disconnectivity analysis may further our understanding of how white matter disconnections may predict tumor growth and recurrence patterns, as well as pre- and post-operative symptoms. Disconnectivity analysis could also inform surgical planning. The Allen Human Brain atlas [53], which integrates gene expression data and neuroanatomical information, has not yet been applied to neuro-oncology. This could potentially represent a powerful tool to investigate normative gene expression for various tumor types. Hypothetically, the results may show the vulnerability of certain brain areas to tumor growth and, possibly, highlight new genetic therapeutic targets. Similarly, sites of recurrence after resection could be assessed based on the normative gene expression of the adjacent brain; once again, these may reveal a pattern of vulnerability of certain brain areas. This could help understand and predict recurrences, as well as optimize surgical timing.

## 4. Practical Framework for Normative Brain Analyses

The use of these advanced normative brain atlases enables the analysis of retrospective patient data using MRI routinely acquired with clinical scans. Two items are necessary to begin: (1) approval from the local ethics board allowing chart review and data consolidation; (2) high spatial resolution MRI (typically a T1 or T2-weighted whole-brain image) allowing the localization of the relevant features (e.g., a tumor) and transformation of the patient data into template brain space (Figure 2). The next step is the identification/segmentation of the feature of interest using the native brain (Figure 2). Most commonly, the images have to be converted from DICOM (i.e., most common output format from MRI scanners) to nifti format [53,54]. There are many publicly available toolboxes that can easily be installed and used to perform these operations, for example SPM (https://www.fil.ion.ucl.ac.uk/spm/, accessed on 1 December 2021), FSL (https://fsl.fmrib.ox.ac.uk/fsl/fslwiki, accessed on 1 December 2021), ANTs (http://picsl.upenn.edu/software/ants/, accessed on 1 December 2021), or MINCtools (https://bic-mni.github.io/, accessed on 1 December 2021). To normalize the images containing the identified/segmented feature of interest (e.g., brain tumor), one would derive linear and non-linear transformations using the native structural MRI and apply these to the image capturing the feature of interest (Figure 2) [11,55,56]. This has to be performed for each individual subject/patient. Once transformed, the first question that can be investigated is whether or not the feature of interest shows a distinct spatial pattern, for example voxels or regions more or less likely to have tumor occurrence (Figure 2).

Using the segmented feature in the template space as the input in the normative functional and structural connectomes, one can then identify the brain-wide network of functionally and structurally connected areas (Figure 2). This allows identification of networks involved in clinical symptoms or disease processes (Figure 1) [22,25,26,27,46]. For example, tumor growth might preferentially move along a specific fiber tract or metastasis might be more likely to appear in brain areas functionally connected.

The spatial pattern of either the segmented feature (i.e., voxel-based analysis) and connectomics analyses can then be related to the spatial pattern of gene expression, which enables investigating the relationship of brain patterns and cell types or molecular processes [53,54,55]. One can use, for example, the regional Allen Gene Expression data (https://human.brain-map.org/, accessed on 1 December 2021) or tools, such as the abagen toolbox [54], to derive gene expression patterns and correlate these with the connectome or tumor occurrence pattern (Figure 1). This allows the identification of the gene set showing similar or opposite expression patterns. Moreover, gene ontology analysis (http://geneontology.org/, accessed on 1 December 2021) can assess processes associated with that gene set. If investigating tumor occurrences, this process could identify biological or molecular processes that make tumor occurrences more or less likely (Figure 1 and Figure 3).

## 5. Limitations of Normative Analyses

While normative analyses could have broad applications to the field of neuro-oncology, it is important to acknowledge their weaknesses. The major limitation of normative datasets is that the templates do not contain information about the brains of the individual patients to which they are applied [5,55]. This means that normative connectomes are not sensitive to intrinsic interindividual differences at a more granular level. Similarly, they may omit certain idiosyncrasies of pathology-specific brain characteristics. Therefore, it is imperative to ask appropriate scientific questions when using normative analyses. The normalization process represents another limitation. Although this has been highly optimized for non-linearly registering subcortical elements with submillimeter precision, normalization remains an imperfect process, particularly when marked morphological changes are present [3]. Of note, certain techniques, including masking specific brain areas or emphasizing certain brain structures during this process, can be used to improve the normalization of specific datasets. Finally, direct electrical stimulation during surgery may still provide more immediate and accurate information for surgical planning than functional and structural imaging, even at the patient level [56]. The unfamiliarity of the neuro-oncology field with these neuroimaging analyses combined with the novelty and requisite expertise to apply these techniques likely accounts for the paucity of neuro-oncology publications taking advantage of normative connectomics and transcriptomics.

These disadvantages are partially offset by considering that normative templates are high-resolution, high-fidelity, and high-n aggregate imaging data from initiatives such as the Brain Genomics Superstruct Project [57,58,59]. This is in contrast with clinically acquired individual patient imaging, which is often limited by suboptimal acquisition parameters due to time constraints and outdated hardware [5,55].

## 6. Conclusions

Normative neuroimaging approaches provide an opportunity to perform large-scale retrospective data analysis with MRI commonly acquired with clinical protocols. Thus far, their usage in neuro-oncology has been limited. We discussed their potential broad novel applications, from assessing tumor characteristics to pre-operative patient counselling and from surgical planning to post-operative treatment monitoring (Figure 3). These analyses may provide different perspectives on the current knowledge base.

## Figures and Tables

**Figure 1 cancers-14-00464-f001:**
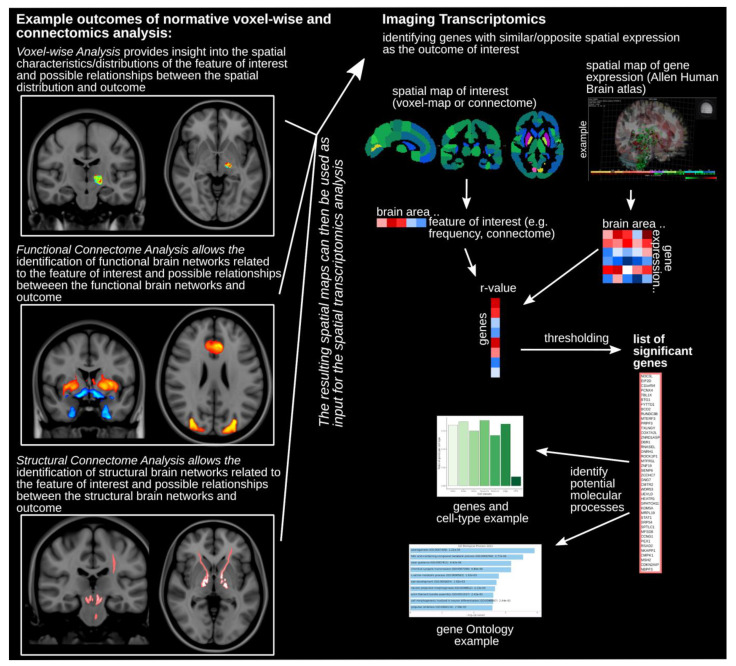
Normative voxel-based, connectomics, and imaging transcriptomics analyses. Examples of outcome maps depicting voxel-based analysis (**top**), functional connectomic analysis (**middle**), and structural connectomic analysis (**bottom**) are shown on the left side of the figure. The resulting spatial maps of those analyses can then be used as input for the spatial transcriptomics analysis outlined on the right side of the figure. Atlas segmentation is used to calculate the correlation of spatial gene expression and imaging results across all brain areas. The thresholded subset of the list identifies the significantly implicated genes. This list can then be further investigated using, for example, gene ontology analysis.

**Figure 2 cancers-14-00464-f002:**
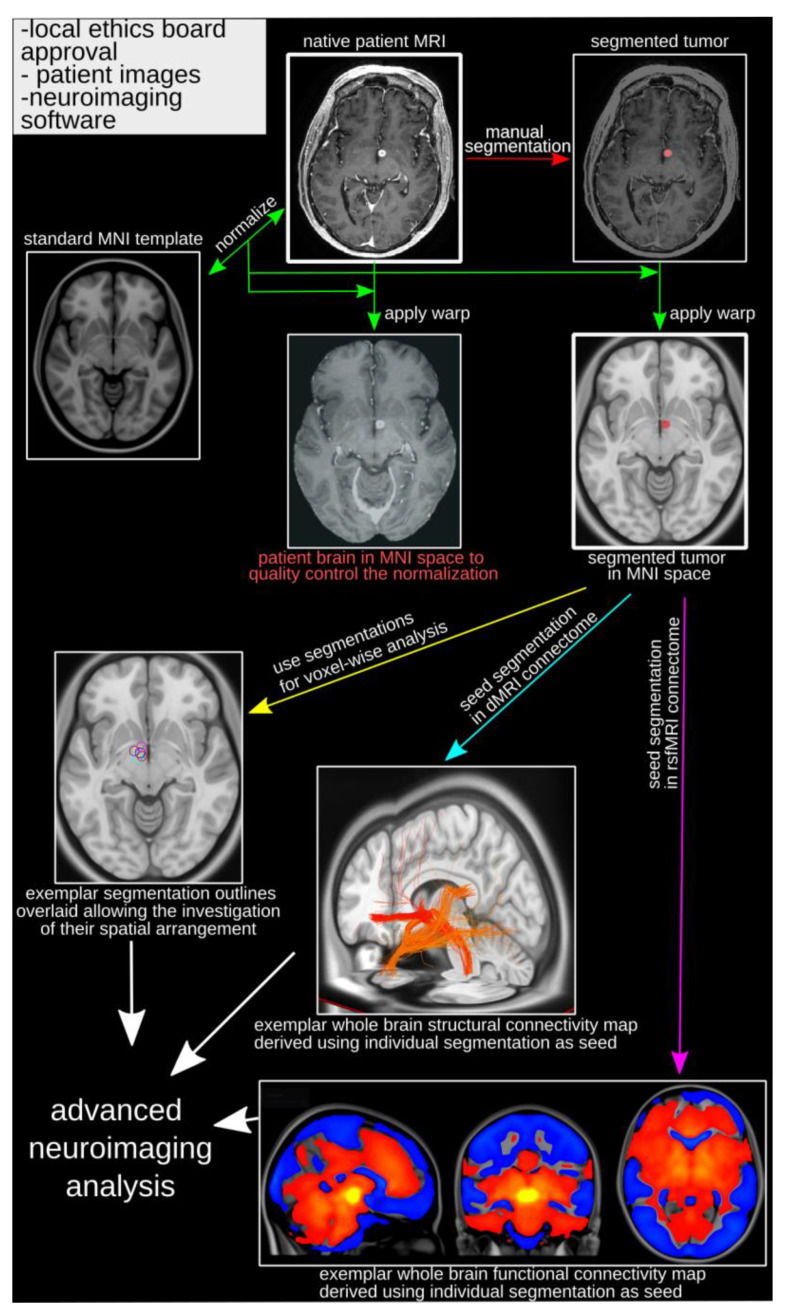
Framework for normative brain analyses. Following typical research project prerequisites (upper left side of the image), the analysis begins with the native patient MRI. The feature of interest (e.g., tumor) is manually segmented (red arrow) using the native patient image. The native patient brain is then normalized (transformed) to MNI space and the estimated transforms applied to the native patient brain (for quality control) and the segmented feature (green arrows). The segmented feature (e.g., tumor) in MNI space is the main input for further processing, such as voxel-based group analysis (yellow arrow), and is used as seeds in normative structural (turquoise arrow) and functional (purple arrow) connectome analyses to derive brain-wide connectivity patterns. dMRI = diffusion-weighted MRI; MNI = Montreal Neurological Institute; MRI = Magnetic resonance imaging; rsfMRI = resting state functional MRI.

**Figure 3 cancers-14-00464-f003:**
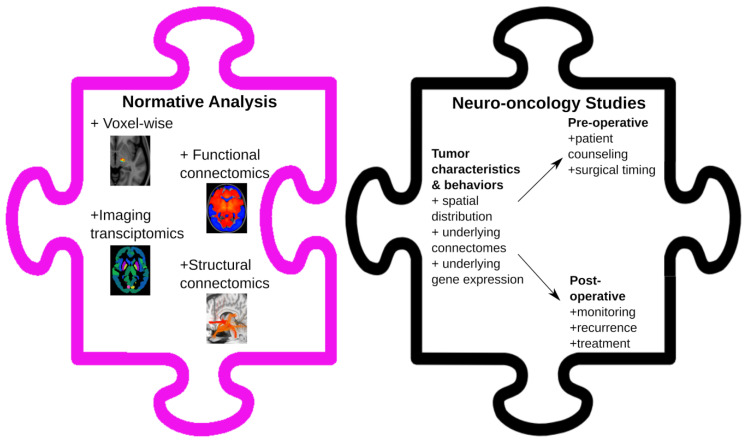
Neuro-oncology applications of normative analyses. Voxel-based, connectomics, and imaging transcriptomics are various tools that can be used when performing normative analyses. Their applications to neuro-oncology are broad and include the assessment of tumor characteristics and behaviors, leading to potential pre- and post-operative improvements for patients.

## Data Availability

The data presented in this study are available on request from the corresponding author.

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
