# Peer review of "Untapped Neuroimaging Tools for Neuro-Oncology: Connectomics and Spatial Transcriptomics"

_cancers, 2022, doi:10.3390/cancers14030464_

Round 1

Reviewer 1 Report

The authors present a very timely review on advanced imaging approaches to brain tumors which evaluate the role of integrating normative analysis, connectomics, and transcriptomics for neuro-oncologic clinical applications. The authors describe approaches of normalizing different patient data into cohesive atlases of data, which allow group analysis. In the first part of the review, the authors describe the methods for how atlases are constructed with detailed and well articulated descriptions. The section on connectomics summarizes how connectome maps are generated. The imaging transcriptomics section describes the Allen Brain Atlas consortium and its potential applications to neuro-oncology. Neuro-oncology applications review the role of atlas based analysis in correlating tumor location to tumor molecular subtype. The authors also describe potential implications to understand how tumors spread in the brain and how they can be helpful in glioma surgery. Practical framework gives a very broad overview of a large field of image processing, which may be appropriate for a general reader but will need more references for readers with different expertise.  The limitations section provides good overview. The authors provide very extensive presentation of the limitations of normative analyses but could more critically deal with the lack of publications on connectomics and transcriptomics in neuro-oncology and why it is not available yet. This will help to advocate for more research in this area. Overall, we are very impressed by the vision of the authors to move the field of neuro-oncology to the next level by providing this insightful review. This review will inspire the readers to focus their research more on evaluation of connectomics and transcriptomics in neuro-oncology.

  • Can you please consider using word ‘atlas’, as opposed to ‘average brain’
  • The figure 1 attempts to represent the overview of the review, but is not clearly explaining the concepts. Specifically, ‘voxel wise analysis’ portion does not explain what the selected voxels are and why they would relate to ‘imaging transcriptomics’ – it would be great to explain what is the point of the 2 ROI selections and how they relate to imaging on the figure. The description in the legend is excellent and it would be great to use the language from the legend in the figure because the way it is now, it is confusing and the message gets lost.  We also recommend to use consistent number of images on the radiologic right for each analysis, because it is not clear why top row has 2 images, middle row has 3 images, and lowest row has 4 images. From neuroradiology standpoint, 3D DTI image (first one) can be excluded and the sag also; coronal and axial CST tracks are enough.  For the rsFMRI images, can also get rid of the sag.  For Voxel wise analysis, it would be great to be consistent and include coronal and axial images. These are just stylistic points for generalizability sake, please feel free to disregard.
  • On page 3, section 2.1: the sentence in line 4 sounds redundant. We recommend “The connectome is key for understanding how the brain works”.
  • The last sentence in the connectomics section needs to be clarified more with an example. The example can be short but will help the reader understand how atlas of connectivity will affect evaluation of a specific patient with a brain tumor in a specific location. Another option is to point the reader toward the upcoming section on ‘neuro-oncology applications’ but that also does not provide specific example.
  • In the imaging transcriptomics section, there is a reference toward application of normative connectomics to Parkinson’s disease. Are there any references on this topic related to neuro-oncology instead?
  • Page 4 second paragraph, spelling error ‘glioblastoma multiformes’, we recommend to actually remove the multiforme and just use glioblastoma according to WHO classification recommendations
  • Practical framework gives a very broad overview of a large field of image processing, which may be appropriate for a general reader. It would be great to reference where reader can get more detailed information depending on the level of the reader. This can be done by making a table of references where more information can be obtained based on level of understanding of image processing. As an example.
  •  

Level of researcher

References from articles

Resources such technical manuals and society based video links if available

Beginner

Some understanding of image processing

Advanced

  • Figure 2: please use uniform spacing and capitalization of labels
  • Figure 3: we appreciate the artistic creativity to drive the point of the two fields of normative analysis and neuro-oncology imaging coming together in harmony like puzzle pieces. We do recommend to use consistent bullets and spacing in the same style as the rest of the figures.  One of the features that may help is small images bc the text appears extensive.  Also, in the caption, please keep same tense ‘their application to neuro-oncology is broad…’ or ‘their applications to neuro-oncology are broad…’

Author Response

Reviewer #1

  1. The authors present a very timely review on advanced imaging approaches to brain tumors which evaluate the role of integrating normative analysis, connectomics, and transcriptomics for neuro-oncologic clinical applications. The authors describe approaches of normalizing different patient data into cohesive atlases of data, which allow group analysis. In the first part of the review, the authors describe the methods for how atlases are constructed with detailed and well articulated descriptions. The section on connectomics summarizes how connectome maps are generated. The imaging transcriptomics section describes the Allen Brain Atlas consortium and its potential applications to neuro-oncology. Neuro-oncology applications review the role of atlas based analysis in correlating tumor location to tumor molecular subtype. The authors also describe potential implications to understand how tumors spread in the brain and how they can be helpful in glioma surgery.

We thank the reviewer for the positive appraisal of our manuscript.

  1. Practical framework gives a very broad overview of a large field of image processing, which may be appropriate for a general reader but will need more references for readers with different expertise.

We thank the reviewer for raising this concern. We have added numerous references in the “Practical framework for normative brain analyses” section to allow easier access to image processing to readers with different expertise. For example, references 11, 22, 25-27, 46, and 53-56 were added.

  1. The limitations section provides good overview. The authors provide very extensive presentation of the limitations of normative analyses but could more critically deal with the lack of publications on connectomics and transcriptomics in neuro-oncology and why it is not available yet. This will help to advocate for more research in this area.

The limitation section was expanded.

The unfamiliarity of the neuro-oncology field with these neuroimaging analyses combined with the novelty and requisite expertise to apply these techniques likely accounts for the paucity of neuro-oncology publications taking advantage of normative connectomics and transcriptomics.

  1. Overall, we are very impressed by the vision of the authors to move the field of neuro-oncology to the next level by providing this insightful review. This review will inspire the readers to focus their research more on evaluation of connectomics and transcriptomics in neuro-oncology.

We appreciate the reviewer’s positive feedback and encouragement.

  1. Can you please consider using word ‘atlas’, as opposed to ‘average brain’

● Generally, the term atlas refers to a set of labels (anatomical, functional, etc.). These atlases are most commonly in normalized/averaged templates/brains for usability. We would prefer keeping the distinct vocabulary as is for accuracy. To prevent further confusion, we modified the title and text of section 2.

2. Advanced neuroimaging analyses using normative brain templates

It serves as the standard reference space for analysis and reporting of results. A multitude of multimodal publicly available atlases have been accurately registered in this standard space.

  1. The figure 1 attempts to represent the overview of the review, but is not clearly explaining the concepts. Specifically, ‘voxel wise analysis’ portion does not explain what the selected voxels are and why they would relate to ‘imaging transcriptomics’ – it would be great to explain what is the point of the 2 ROI selections and how they relate to imaging on the figure. The description in the legend is excellent and it would be great to use the language from the legend in the figure because the way it is now, it is confusing and the message gets lost.

We apologise for the lack of clarity in figure 1. We have altered the layout and the text of the figure to clarify the concept illustrated. As suggested, we have taken some of the language from the figure legend. We hope the new figure (see below) illustrates our concepts and ideas better.

  1. We also recommend to use consistent number of images on the radiologic right for each analysis, because it is not clear why top row has 2 images, middle row has 3 images, and lowest row has 4 images. From neuroradiology standpoint, 3D DTI image (first one) can be excluded and the sag also; coronal and axial CST tracks are enough. For the rsFMRI images, can also get rid of the sag. For Voxel wise analysis, it would be great to be consistent and include coronal and axial images. These are just stylistic points for generalizability sake, please feel free to disregard.

We thank the reviewer for pointing out this inconsistency. We have changed figure 1 and now use a coronal and axial image for each illustration (see new figure 1 in the previous answer).

  1. On page 3, section 2.1: the sentence in line 4 sounds redundant. We recommend “The connectome is key for understanding how the brain works”.

● Thank you for the suggestion. The sentence was changed accordingly.

  1. The last sentence in the connectomics section needs to be clarified more with an example. The example can be short but will help the reader understand how atlas of connectivity will affect evaluation of a specific patient with a brain tumor in a specific location. Another option is to point the reader toward the upcoming section on ‘neuro-oncology applications’ but that also does not provide specific example.

● We appreciate the concern and, following the suggestion, have added an example at the end of the paragraph.

For example, if tumors are found to cause a specific cognitive impairment, one could use the normative functional connectome to map the key implicated regions and optimise patient counselling and surgical timing of subsequent tumors with similar characteristics.

  1. In the imaging transcriptomics section, there is a reference toward application of normative connectomics to Parkinson’s disease. Are there any references on this topic related to neuro-oncology instead?

● We thank the reviewer for pointing out this omission. We have added a neuro-oncology example of Mandal et al. (2020).

For example, a study by Zheng and colleagues used normative connectomics and transcriptomics to investigate determinants of disease propagation in Parkinson's disease [41] and recent work by Mandal and colleagues demonstrated that spatial distribution of gliomas is related to the local normative expression of genes associated with chromatin organisation and synaptic signalling. [42]

  1. Page 4 second paragraph, spelling error ‘glioblastoma multiformes’, we recommend to actually remove the multiforme and just use glioblastoma according to WHO classification recommendations.

● This was changed accordingly.

  1. Practical framework gives a very broad overview of a large field of image processing, which may be appropriate for a general reader. It would be great to reference where reader can get more detailed information depending on the level of the reader. This can be done by making a table of references where more information can be obtained based on level of understanding of image processing. As an example.

● We thank the reviewer for raising this concern. As mentioned before, we have added numerous references in the “Practical framework for normative brain analyses” section. These together with the additional references added throughout the text will provide starting points for readers to obtain more detailed information. Furthermore, there are manuals and detailed instructions that accompany the imaging tools mentioned, which provide a good starting point for the readers. As the original goal of the manuscript was to provide a general overview of how these techniques could be applied to the field of neuro-oncology, a more detailed overview of the various imaging techniques is unfortunately outside the scope of this manuscript.

  1. Figure 2: please use uniform spacing and capitalization of labels

We have made the spacing and the labels more uniform. We thank the reviewer for the suggestion.

  1. Figure 3: we appreciate the artistic creativity to drive the point of the two fields of normative analysis and neuro-oncology imaging coming together in harmony like puzzle pieces. We do recommend to use consistent bullets and spacing in the same style as the rest of the figures. One of the features that may help is small images bc the text appears extensive. Also, in the caption, please keep same tense ‘their application to neuro-oncology is broad…’ or ‘their applications to neuro-oncology are broad…’

● We thank the reviewer for these remarks. Our new and improved figure with small images - as per the reviewer’s suggestions - is illustrated below.

Reviewer 2 Report

This perspective paper is focused on the possibilities provided by the analysis of groups of structural brain MRI in neurooncology, including  investigations of underlying connectomics thanks to functional and structural templates and, assessment of associated gene expression. These possibilities largely unexplored and could represent an opportunity to address neuro-oncology questions from a different perspective.

This manuscript is well written and deals a new and original concept that could be very promising.

I have very few remarks.

  1. Some limitations are represented by the use of functional MRI which do not perfectly correlate with the data obtained by electrical brain mapping (D’Azad et al, Neurosurg focus, 2020) and may thus introduce bias.
  2. I am a little bit sceptic regarding transcriptomics. Evidently, the intertumoral and intratumoral transcriptomic heterogeneity is huge. How will it be possible to deal with this limitation ?

Author Response

Reviewer #2

  1. This perspective paper is focused on the possibilities provided by the analysis of groups of structural brain MRI in neurooncology, including investigations of underlying connectomics thanks to functional and structural templates and, assessment of associated gene expression. These possibilities largely unexplored and could represent an opportunity to address neuro-oncology questions from a different perspective.

This manuscript is well written and deals a new and original concept that could be very promising.

We thank the reviewer for the positive appraisal of our manuscript.

  1. I have very few remarks.

Some limitations are represented by the use of functional MRI which do not perfectly correlate with the data obtained by electrical brain mapping (D’Azad et al, Neurosurg focus, 2020) and may thus introduce bias.

● We thank the reviewer for pointing out this important point. We have added a sentence to the limitations sections to address this.

Finally, direct electrical stimulation during surgery may still provide more immediate and accurate information for surgical planning than functional and structural imaging, even at the patient level [56].

  1. I am a little bit sceptic regarding transcriptomics. Evidently, the intertumoral and intratumoral transcriptomic heterogeneity is huge. How will it be possible to deal with this limitation ?

● We agree with the reviewer that the tremendous heterogeneity is one of the limiting factors of using normative transcriptomics for oncology studies. Nevertheless, Mandal and colleagues in their recent work (doi:10.1093/brain/awaa277, now cited as an example) show - using the normative Allen human brain gene atlas - that spatial gene expression patterns combined with connectomics is able to explain a considerable portion (58%) of the spatial variance in glioma frequency. Thus, we assume that a non-trivial amount of interindividual variance in oncology patterns will be able to be attributed to “typical” gene expression patterns using the methods proposed in our work. Finally, we think that this technique could provide new insights to the field of neuro-oncology as the underlying brain characteristics may play a role in tumor development that is not easily captured with more “traditional” research using tumor biopsy data.
